# Accountable Textual-Visual Chat Learns to Reject Human Instructions in Image Re-creation

**Zhiwei Zhang**                                                                     *bitzzw@gmail.com*
*The Chinese University of Hong Kong*
*Centre for Perceptual and Interactive Intelligence*

**Yuliang Liu**[†]                                                                   *ylliu@hust.edu.cn*
*The Chinese University of Hong Kong*
*Huazhong University of Science and Technology*
*Centre for Perceptual and Interactive Intelligence*

**Reviewed on OpenReview:** *https://openreview.net/forum?id=kQmz1BMIYi*

## Abstract

The recent success of ChatGPT and GPT-4 has drawn widespread attention to multimodal dialogue systems. However, there is a lack of datasets in the academic community that can effectively evaluate the multimodal generation capabilities of Visual Language Models (VLMs) in textual-visual chat tasks. In this paper, we address this gap by introducing two novel multimodal datasets: the synthetic CLEVR-ATVC dataset (620K) and the manually pictured Fruit-ATVC dataset (50K). These datasets incorporate both visual and text-based inputs and outputs. Furthermore, to facilitate the accountability of multimodal systems in rejecting human requests, similar to language-based ChatGPT conversations, we introduce specific rules as supervisory signals within the datasets. This allows the trained VLM to provide a yes or no answer after engaging in visual and textual reasoning, accompanied by a language explanation to clarify the reasons behind the inability to execute the given human instruction. Our proposed method involves a two-stage training procedure, which includes training the image auto-encoder and the auto-regressive transformer from scratch. The first stage employs a discrete variational autoencoder (dVAE) to compress each image into concise tokens, which are then combined with text tokens into a single data stream. This stream is subsequently fed into the decoder-based transformer to generate visual re-creations and textual feedback in the second stage. We conduct comprehensive analyses of experimental results, focusing on re-created image quality, answer accuracy, and the model's behavior when faced with uncertainty and imperfect user queries. Through our explorations and findings, we aim to contribute valuable insights into the accountability of textual-visual generative models. Dataset and code are available at https://matrix-alpha.github.io.

## 1 Introduction

Recently, the most important breakthrough was made by ChatGPT (OpenAI, 2023a) and GPT-4 (OpenAI, 2023b), which unveiled the emerging potential of the conversation between human and artificial intelligence system. ChatGPT serves as a chatbot that operates with language as both input and output, while GPT-4 is a multimodal model capable of accepting both image and text inputs and producing text outputs. A successful multimodal generative model should excel in both textual and visual reasoning, generating high-quality text and image feedback. Visual ChatGPT (Chenfei Wu & Duan, 2023) is a pioneering work that combines ChatGPT with a series of pre-trained visual foundation models, enabling text-image chat. Another relevant work, FROMAGe (Jing Yu Koh, 2023), also involves image-text inputs and outputs for

---

[†]Corresponding author: Yuliang Liu.

multimodal dialogue, with the fine-tuning of linear layers and the freezing of pre-trained LLM. However, existing datasets lack definitive ground truths to effectively measure the quality of text and images generated in multimodal dialogue systems. Therefore, there is an urgent need for a dataset that can evaluate the performance of multimodal generative models. In this paper, we aim to address this need by constructing new multimodal datasets that require models to output high-quality images and provide textual feedback while accepting a text-image pair. We introduce the synthetic CLEVR-ATVC dataset (620K), created by rendering images using approximately 200 GPU days, and the real Fruit-ATVC dataset (50K), manually captured and annotated.

Another significant contribution of this paper pertains to the issue of responsibility in text-to-image generation models, specifically the need for models to learn how to reject human instructions. Previous works (Doshi-Velez et al., 2017; Loi & Spielkamp, 2021) have highlighted the importance of accountability in AI systems, including the ability to hold decision-makers accountable for adhering to procedural and substantive standards. The responsible management and deployment of AI systems is a crucial and often overlooked topic. ChatGPT (OpenAI, 2023a) implements a similar requirement at deployment time by employing rules defined by OpenAI to restrict certain requests. However, to the best of our knowledge, we are the first to consider the issue of responsibility in textual-visual dialogue models. Previous text-to-image generation methods (Ramesh et al., 2021a; Nichol et al., 2021; Wu et al., 2021; Ramesh et al., 2022; Saharia et al., 2022; Ding et al., 2021) have produced random and uncontrolled outcomes lacking human oversight. While some works have explored the controllability of image generation by focusing on object categories or attributes (e.g., position, pose, color) (Li et al., 2019; Niemeyer & Geiger, 2021; Jiang et al., 2022; Patashnik et al., 2021), such as StyleCLIP (Patashnik et al., 2021), which manipulates visual image attributes using text, the rejection of human instructions and providing explanations for such decisions have not been adequately addressed in the textual-visual dialogue context. Therefore, in this paper, we propose that the machine should be capable of rejecting human instructions and providing explanations, particularly for instructions that cannot be executed or are prohibited.

To serve the aforementioned research goals, we introduce a novel task called accountable text-based visual re-creation, which explores whether textual-visual chat models can learn to reject human instructions. This task requires the model to generate both visual re-creations and language-based feedback while accepting a text-image pair. Our datasets include rules as supervision signals to teach the model to reject certain instructions. As illustrated in Figure 1, when a chatbot receives instructions from a human, it responds with a "no" to forbidden or unfeasible instructions, accompanied by an explanation. An instruction may be deemed unfeasible if the corresponding objects mentioned in the text-based query are absent in the visual input. Prohibited instructions are determined manually, taking into account the laws of physics. Prohibited actions encompass both actions that are feasible but not allowed, as well as actions that are simply not possible. In the case of prohibited instructions that cannot be executed, the model should provide additional explanations to clarify the reasons behind their infeasibility. These instructions are elaborated and listed in Table 1. Our proposed method, as depicted in Figure 1, employs a multimodal generative model comprising an image auto-encoder and an auto-regressive transformer. We propose a two-stage training procedure to train these components from scratch. The first stage involves a discrete variational autoencoder (dVAE) that compresses each image into concise tokens, which are then concatenated with text tokens and fed into the decoder-based transformer responsible for generating the visual re-creations and textual feedback.

Additionally, we provide comprehensive analysis of our experimental results, including assessments of re-created image quality, answer accuracy, and model's behavior when faced with uncertainty and imperfect user queries. We also compare two different image auto-encoders, VQVAE (Oord et al., 2017) and VQGAN (Esser et al., 2021), and analyze the reasons behind the subpar performance of VQGAN. All the datasets used in this study are publicly available, and we will release the source code for our annotation tool, evaluation tool, implementation of baseline models, metric calculations, and detailed instructions.

To summarize, the main contributions of this paper are as follows:

- We construct two multimodal datasets, CLEVR-ATVC (620K) and Fruit-ATVC (50K), which incorporate textual-visual inputs and outputs to evaluate the performance of multimodal generative models.

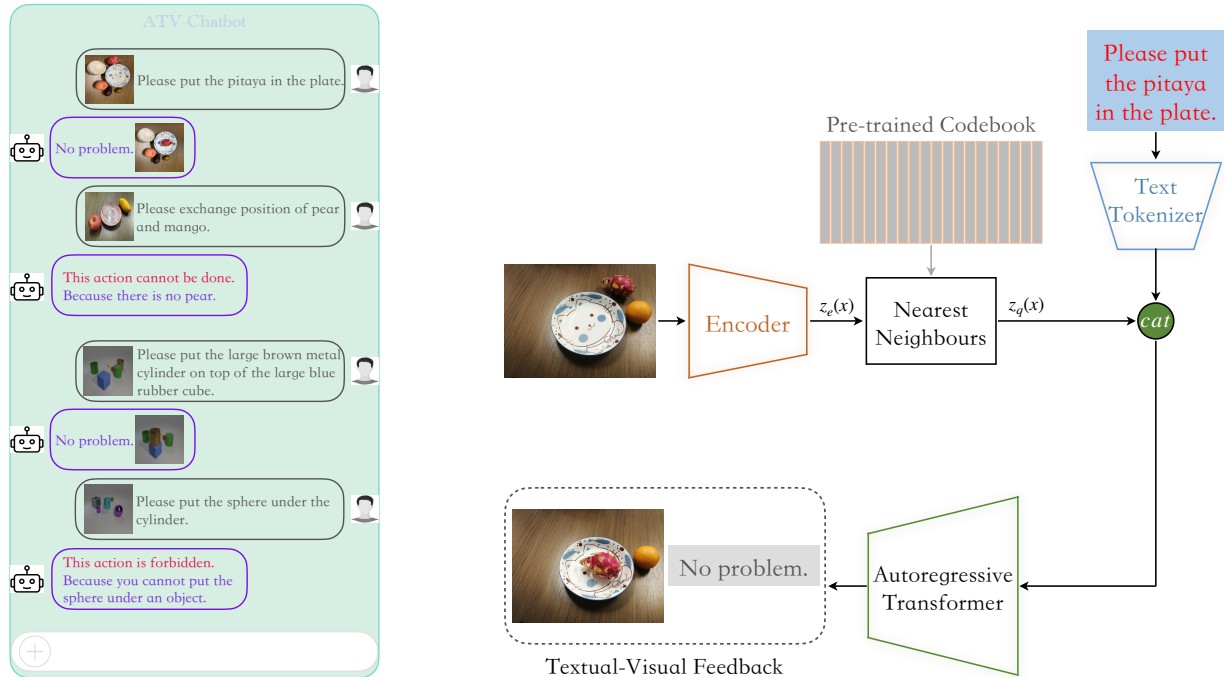

Figure 1: The left figure shows that a multimodal generative model can simultaneously recreate visual input and give textual feedback when faced with human instructions, especially it can reject some commands. The right figure illustrates the overall framework of our method. The model is required to generate a re-created image ($\mathbf{M}$) and a textual feedback ($\mathbf{A}$) conditioned on the visual input ($\mathbf{V}$) and text-based user query ($\mathbf{T}$), and the language-based explanation is also given for those instructions that cannot be executed and the prohibited instructions.

- We consider the issue of accountability in multimodal generative models by embedding pre-set rules as supervised signals in our datasets. This enables the VLMs to learn to reject human instructions in multimodal conversations.

- We propose a two-stage training procedure for training the image auto-encoder and auto-regressive transformer, aiming to enable the models to learn how to reject human instructions. All of our models are trained from scratch, and their training took 350~900 GPU days on our constructed datasets.

- We provide extensive qualitative and quantitative results, evaluating the quality of generated images, the accuracy of answers, and the model's ability to handle uncertainty and incomplete queries.

The remainder of this paper is organized as follows. Section 2 presents detailed information about our datasets. In Section 3, we introduce our models and training process. We then present extensive experiments and detailed analysis in Section 4 to validate our proposed method. Section 5 provides an overview of related work, and Section 6 concludes the paper. Section 7 pertains to acknowledgments. Additional details of our datasets are provided in the appendix.

## 2 Datasets

In this section, we present the distinctive aspects of our datasets compared to existing ones. We then describe the data collection process. Additionally, we provide an overview of the information contained in the annotation files for our datasets, with more detailed information available in the appendix. Finally, we summarize the dataset statistics in the tables.

### 2.1 Definition of ATVC

Our proposed dataset and task introduce two key innovations compared to previous methods. Firstly, our multimodal dataset comprises image-text inputs and outputs, which facilitates the development of more effective multimodal generative models. Secondly, our datasets incorporate predefined rules that teach the model to reject certain human instructions, even if they are technically feasible. This gives rise to the concept of accountable text-based visual re-creation (ATVC) task, which can be defined as follows:

Given a visual input ($\mathbf{V}$) and a text-based query ($\mathbf{T}$), the multimodal generative model is expected to produce a re-created image ($\mathbf{M}$) and provide a language-based explanation ($\mathbf{A}$) for its decisions. A successful ATVC model should possess the ability to comprehend the user's query and employ language cues to reason about the visual input, while also generating appropriate textual feedback and re-created visual results.

### 2.2 Data Collection

In order to validate the proposal of this paper, we conduct experiments on two distinct datasets. The first dataset, CLEVR-ATVC, is a large-scale synthetic dataset specifically designed to comprehensively evaluate the proposed task. On the other hand, the Fruit-ATVC dataset is utilized to evaluate our method on real-world scenarios. In the following sections, we provide a detailed description of the visual image collection process for both datasets.

**CLEVR-ATVC:** We utilize the default environmental settings, including object settings and rendering options, from CLEVR (Liu et al., 2019) to generate synthetic images. The rendered images have a size of 256, with each image containing 3 to 6 objects. In addition, we apply a desiged algorithmic filter to exclude scenes with objects located on the image border before rendering. This ensures that all objects in both the original and re-created images are within the boundaries, facilitating the evaluation of experimental results. The synthetic CLEVR-ATVC dataset is generated over approximately 200 GPU days.

**Fruit-ATVC:** This dataset consists of real-world scenarios where all the images are captured manually by the authors using mobile phones and tripods. The images are taken with various mobile phone models, including iPhone, Huawei, Xiaomi, and others. For the Fruit-ATVC dataset, five researchers contributed to capturing the visual inputs and re-created images. Additionally, we incorporated 12 different scenes to enhance the diversity of the dataset. The diversity of collection scenes is illustrated in the figures provided in the appendix.

### 2.3 Generation of Textual Query and Feedback

For the CLEVR-ATVC dataset, we employ an automated program that simultaneously generates visual inputs, text-based queries, re-created images, and textual feedbacks. This program saves the visual inputs and re-created images in their respective folders, while the text-based queries, textual feedbacks, and other data information are automatically stored in an annotation file. Each visual input is associated with ten different text-based queries and their corresponding textual feedbacks. Among these queries, six instructions can be executed by the AI system, resulting in appropriate image re-creations, while two instructions cannot be executed due to the presence of objects mentioned in the text-based queries that are not present in the visual input. Additionally, two instructions are deliberately prohibited by manual specification. The number of each instruction is approximately the above ratio, because some visual inputs cannot generate prohibited instructions.

The process of generating text-based queries and textual feedbacks for the Fruit-ATVC dataset differs from the aforementioned process. Since the visual inputs and re-created images are manually captured, and the text-based queries and textual feedbacks are generated semi-automatically using our designed labeling tool. Initially, we collect and organize the visual inputs and re-created images according to predefined rules, ensuring the inclusion of executable and non-executable queries for each visual input. Then, we utilize an annotation interface (illustrated in Figure 2) to assist in generating text-based queries and textual feedbacks, which are automatically saved to an annotation file. The specific process is as follows: we select image pairs consisting of the visual input and the re-created image for executable instructions. These image pairs are

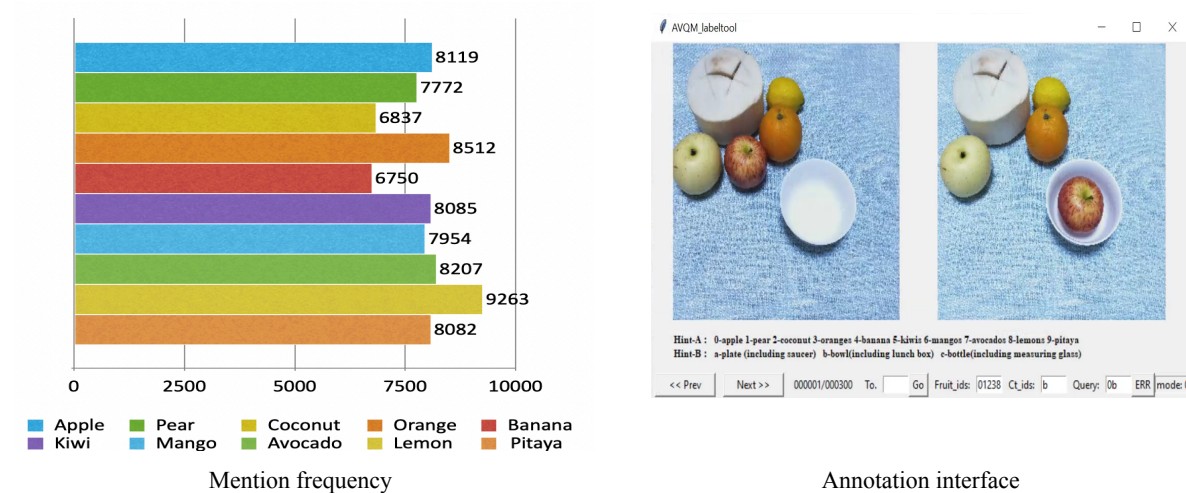

Mention frequency                                    Annotation interface

Figure 2: The mention frequency of different fruits and the annotation interface for Fruit-ATVC dataset.

Table 1: The detailed information of textual feedback.

| Textual Feedback | | |
|---|---|---|
| Answer Type | Reason for prohibition | Explanation |
| 1) No problem. | / | / |
| 2) Cannot be done. | / | no mentioned objects |
| 3) Forbidden. | cannot put certain object under an object | no mentioned objects |
| 4) Forbidden. | cannot put an object on top of certain object | no mentioned objects |

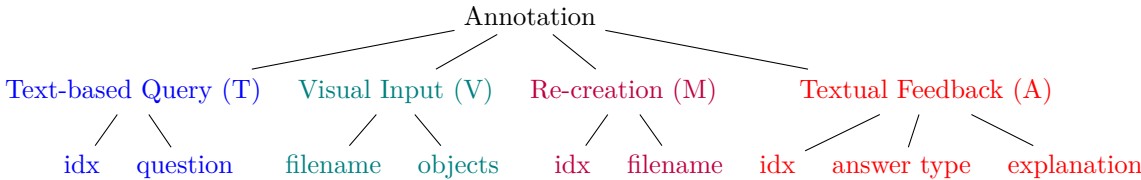

Figure 3: The main format visualization of data annotation file. The details can be found in the appendix.

then imported automatically through the interface depicted in Figure 2, where the object ID, container ID, and query type are manually added to the designated fields on the interface ("fruit_ids", "Ct_ids", and "Query", respectively). The "fruit_ids" field includes all the fruit types depicted in the image, while the "Query" field is filled with the object and container categories involved in the operation. For instance, "0b" represents an apple and a bowl. Finally, by pressing the "ERR" key, text queries and feedback are automatically generated.

## 2.4 Data Annotation

In this section, we present the structure of our data annotation file, which follows the format of the COCO dataset (Lin et al., 2014). The main format, depicted in Figure 3, consists of four parts: text-based query (T), visual input (V), re-created image (M), and textual feedback (A). Since each visual input corresponds to multiple text-based queries, we utilize an "idx" value to establish a one-to-one mapping between T, M, and A. The "objects" subkey enumerates the objects present in the visual input. The "explanation" provides a rationale for prohibition and explains the absence of certain objects. Table 1 provides further details on the textual feedback. There are three types of answers: a) no problem; b) the action cannot be performed; and c) the action is prohibited. Two specific instructions are prohibited: placing a certain object under another object and placing an object on top of a certain object. The prohibited actions include those that can be

Table 2: The summary information of CLEVR-ATVC dataset.

| CLEVR-ATVC | | | | | |
|---|---|---|---|---|---|
| Size | Train Pairs | Test size | Textual-Visual Feedback | Data type | |
| 68 GB | 619,741 | 5,000 | Yes | Synthetic | |
| Please <action1> {size color material} [object] <action1> {size color material} [object]. | | | | | |
| Please <action2> of {size color material} [object] and {size color material} [object]. | | | | | |
| Action1 | Action2 | Size | Color | Material | Object |
| put on top | exchange color | big | red, green, blue, cyan | metal | cylinder |
| put under | exchange position | small | purple, brown, yellow | rubber | cube, sphere |

Table 3: The summary information of Fruit-ATVC dataset.

| Fruit-ATVC | | | | |
|---|---|---|---|---|
| Size | Train Pairs | Test size | Textual-Visual Feedback | Data type |
| 168 GB | 27,503 | 1,872 | Yes | Real |
| Please <action1> [object]. | | | | |
| Please <action2> of [object] and [object]. | | | | |
| Please <action3> [object] <action3> [container]. | | | | |
| Action1 | Action2 | Action3 | Object | Container |
| remove | exchange position | put in | apple, coconut, lemon etc. | plate, bottle, etc. |

performed and those that cannot be performed. For prohibited instructions that cannot be executed, the model needs to provide further explanations. Please refer to the example given in Table 4. In addition, the results in Table 6 show that we conducted statistics on accuracy of answer type and accuracy of explanation in textual feedback.

## 2.5  Data Statistics

Table 2 and Table 3 present the summarized information of our constructed datasets.

**CLEVR-ATVC** consists of a training set with a total of 620,000 pairs and a testing set with 500 visual inputs, each accompanied by 10 queries, resulting in a total of 5,000 pairs. This dataset incorporates four action types: "exchange color", "exchange position", "put under", and "put on top". The rendered images encompass three object shapes, seven object colors, two object materials, and other attributes, leading to hundreds of millions of possible combinations. The text-based queries follow two different templates: the first template is used for "put on top" and "put under" actions, while the second template is employed for "exchange color" and "exchange position" actions.

**Fruit-ATVC** comprises 27,503 training pairs and 1,872 pairs in the testing set. This dataset involves ten different types of fruits, namely apple, pear, coconut, orange, banana, kiwi, mango, avocado, lemon, and pitaya. The frequency of mentions for each fruit is depicted in Figure 2. Additionally, the dataset includes 14 types of containers. Regarding the queries, there are three types of actions: "put a fruit in a container", "exchange positions of the fruits", and "remove the container".

## 3  Method

In our method, the generative model includes an image auto-encoder and an auto-regressive transformer (Vaswani et al., 2017) to predict tokens of re-created image and textual feedback. The framework of our

method is shown in Figure 1. Our method is a two-stage training procedure, similar to Ramesh et al. (2021a).

**Image Auto-encoder** This stage forces the model to learn the sequential discretized latent space of high dimensional images to utilize the expressive transformer architecture, *i.e.*, learning a codebook to encode the visual input. Given an image of resolution $256 \times 256$, a quantization model encodes it into $32 \times 32 = 1024$ discretized latent codes where the codebook size is 8192. We have tried two alternatives, including Vector Quantized Variational Autoencoder (VQVAE) (Oord et al., 2017) and VQGAN (Esser et al., 2021). As illustrated in Figure 1, the VQVAE can learn a latent space matrix with discrete learnable variables, which is trained end-to-end via straight-through estimation. The codebook and the VQVAE model can be trained end-to-end using the following objective:

$$\mathcal{L}_{VQ}(G,C,E) = \underbrace{\|sg[E(v)] - e\|_2^2}_{\mathcal{L}_{codebook}} + \underbrace{\beta \|sg[e] - E(v)\|_2^2}_{\mathcal{L}_{commit}} \\ + \underbrace{\|v - G(e)\|_2^2}_{\mathcal{L}_{rec}}, \tag{1}$$

where $v$ is the visual input, $e$ is the embedding for learning the codebook-indices, $C$ is the learned codebook, $E$ is the image encoder, $G(e)$ is the decoder for reconstruction. $sg$ is the stop-gradient operation, and $\beta$ is a hyper-parameter to balance the commitment loss (Oord et al., 2017). The quantized codebook index is determined by looking up the nearest codebook vector of input features $E(v)$ in terms of the Euclidean distance. The VQGAN (Esser et al., 2021) is a variant of VQVAE, which adds a perceptual loss and an adversarial loss produced by patch-based discriminator. It can achieve high resolution while significantly reducing the sequence length. The complete objective for the compression model $Q^*$ is:

$$Q^* = \arg\min_{G,C,E} \max_{D} \mathbb{E}_{x \sim p(v)}[\mathcal{L}_{VQ}(G,C,E) \\ + \lambda \mathcal{L}_{GAN}(\{G,C,E\},D)], \tag{2}$$

where $\mathcal{L}_{GAN}$ is to learn the differences between real and reconstructed images:

$$\mathcal{L}_{GAN}(\{G,C,E\},D) = [\log D(v) + \log(1 - D(\hat{v}))]. \tag{3}$$

**Auto-regressive Transformer** Based on the first stage, the images can be highly compressed by the codebook-indices of their embeddings. In this state, the image encoder $E$ is fixed. Therefore, the **V**, **T**, **M**, and **A** can be represented by the embedding $\mathbb{S}$ of the sequence of tokens. We adopt the axial positional embedding $\mathbb{P}$ to process the codebook-indices generating by the image reconstruction module $\mathbb{R}$, which is practically effective for multi-dimensional data. In our implementation, the sequence $T_{seq}$ of the transformer is sequentially formed by **T**, **V**, **M**, and **A**, which is defined as follows:

$$T_{seq} = \text{Concat}(\mathbb{S}(\mathbb{O}(T)), \mathbb{P}(\mathbb{R}(V)), \mathbb{P}(\mathbb{R}(M)), \mathbb{S}(\mathbb{O}(A))), \tag{4}$$

which focuses to generate the re-created result and textual feedback. $\mathbb{O}$ represents the tokenize operation. The transformer is designed as a decoder-only model, which is same as Child et al. (2019). In our architecture, text tokens have the ability to attend to each image token in any of the self-attention layers. The model employs two types of self-attention masks. The attention masks used for text-to-text attention follow the conventional causal mask. On the other hand, for image-to-image attention, the model applies either a row, column, or convolutional attention mask in different self-attention layer, as explained in Child et al. (2019); Ramesh et al. (2021a). During training, each token is asked to predict the distribution of the next token, which formulates as an autoregressive next-index prediction. Therefore, it is equivalent to maximize the log-likelihood of the data representations.

$$\mathcal{L}_{Transformer} = \mathbb{E}_{x \sim p(x)}[-\log p(T_{seq})], \tag{5}$$

where $p$ is the full representation of the likelihood of the possible next indices. Note that for the "cannot" and "forbidden" pairs, as there is no re-created ground truth, the loss of the its prediction will not be back-propagated. We have tried to predict a black image or the original visual input for these two cases, but both of them perform much worse than simply ignoring their loss. In the testing procedure, we only need to input the **T** and **V** for pixel-wise iterative generation until the required length is achieved.

## 4 Experiment

In this section, we introduce the experimental details, evaluation metrics, and analyse on the quantitative and qualitative results.

### 4.1 Experimental Details

All experiments are all conducted with Pytorch 1.8.1. The model parameters are updated through Adam (Diederik P. Kingma, 2014) with $\beta_1 = 0.9$, $\beta_2 = 0.999$. We first trains the image reconstruction module for 200 epochs with an initial learning rate of 0.001 and a decay rate of 0.99 per epoch. This stage takes about half a day to train for 200 epochs. The number of attention heads, the attention key and value dimensions, the number of layers, and the dimensions of the models are set to 8, 64, 4, and 512, respectively. The text tokenizer operates on a lower-cased byte pair encoding (BPE) representation of the text with a 49,408 vocab size (Sennrich R & A, 2015). The text sequence is bracketed with [SOS] and [EOS] tokens. The maximum length of the text sequence is set to 64, and the output length of the image codebook is 1024, which results in a total 2112 sequence lengths for the transformer model. The second stage is distributively trained over 200 epochs with a fixed learning rate of 0.0003. Unless specified, all the parameters of the architecture for the following experiments will remain the same. The model is trained for approximate 900 GPU days on CLEVR-ATVC dataset and 350 GPU days on Fruit-ATVC dataset.

### 4.2 Evaluation Metrics

We evaluate the proposed method using the following two types of metrics.

**Image Quality Metric.** The first two are the Peak signal-to-noise ratio (PSNR) and the structural similarity index measure (SSIM), which are commonly used to measure the similarity and quantify reconstruction quality of images. We also adopt the FSIM metric Zhang et al. (2011), which could be more suitable for the human visual system (HVS) using phase congruency (PC) and gradient magnitude (GM).

**Human Evaluation.** Following the previous methods (Lee et al., 2020; Ding et al., 2021; Dong et al., 2017; Zhang et al., 2017; 2018; 2020; Kayser et al., 2021), we also adopt Human Rank (HR) to precisely reveal the performance. HR can be used to evaluate whether the synthesized image conforms to subjective effects (authenticity, matching degree with text, etc.). We set three different classes for the HR: a) if the action is perfectly done, the result would be ranked "A", representing score 1; b) if the action is correct, but other parts are affected, *e.g.*, mistakenly change the color or erasing other irrelevant objects, the result would be ranked "B", representing score 0.5; c) if the action is incorrect, the result is "C", representing score 0. Therefore, the final HR score is between 0 and 1, and the Full-Match score (FM-Score) represents that both the re-creation and the answer are correct. The higher scores of all metrics, the better performance is represented. The "accountable" part is totally automatic, for which we use the full string matching to measure the accuracy between ground truth answers and the final predictions. We also take into account that the textual feedback in the explanation part will change, such as that both explanations "There is no object1 and object2" and "There is no object2 and object1" are correct. We also designed tools to help with human evaluation, and we provide its interface and evaluation criteria in the appendix. In addition, we assign 100 sets of results to 5 works for scoring, we empirically find that our task usually has an explicit answer which is not easily affected by human subjective opinions.

### 4.3 Results and Analyses

In this section, we evaluate the proposed method on the CLEVR-ATVC and Fruit-ATVC datasets, whose results include the image re-creation, textual feedback, qualitative visualizations and the model behavior when facing uncertainty.

#### 4.3.1 Results on CLEVR-ATVC

For the CLEVR-ATVC dataset, the re-created results are shown in Table 5. We can see that the machine achieves FM-Score 52.6% (re-creation and textual feedback are completely correct), which also shows the

Table 4: Qualitative results of CLEVR-ATVC dataset.

| Visual (V) | Text (T) | Re-created (M) | Answer (A) | **FM-Score** |
|---|---|---|---|---|
|  | Please exchange the positions of the large brown metal cylinder and the large blue rubber cube. |  | No problem. | **1.0** |
|  | Please put the large brown metal cylinder on top of the large blue rubber cube. |  | No problem. | **0.75** |
|  | Please put the small gray rubber cylinder on top of the large yellow metal cube. | / | This action cannot be done. Because there is no large yellow metal cube. | **1.0** |
|  | Please put the small gray metal sphere under the small purple rubber cylinder. | / | This action is forbidden. Because you cannot put the sphere under an object, and there is no small gray metal sphere and no small purple rubber cylinder. | **1.0** |

Table 5: Image re-creation evaluation on the CLEVR-ATVC dataset.

| PSNR | SSIM | FSIM | Human Rank (%) | | | | |
|---|---|---|---|---|---|---|---|
| | | | A | B | C | Score | FM-Score |
| 55.0 | 0.9944 | 0.669 | 25.9 | 16.2 | 58.0 | 39.0 | 52.6 |

Table 6: Texual feedback evaluation on the CLEVR-ATVC dataset. Type Acc. represents that the answer can correctly recognize the query that cannot be done or forbidden. Exp Acc. represents the explanation is correct in both answer type and the reasons.

| Can | | Cannot | | | Forbidden | | | Score (%) |
|---|---|---|---|---|---|---|---|---|
| Num | Acc. | Num | Type Acc. | Exp Acc. | Num | Type Acc. | Exp Acc. | |
| 1662 | 71.6% | 1997 | 49.4% | 46.5% | 1341 | 99.7% | 58.2% | 66.2 |

difficulty of the task. The high SSIM 0.9944 represents the model can obtain high image reconstruction quality. Based on our observation, we find that the "exchange color" queries are mostly correct, while "exchange positions" queries are somewhat limited by the image reconstruction.

The quantitative results of textual feedback are shown in Table 6. We can see that there are 1662 pairs for the results on "can" queries; among them, 71.6% pairs are answered correctly. For the queries about "cannot", 49.4% pairs can be accurately recognized that the query is not doable, and 46.5% pairs can not only answer the type but correctly answer the reasons. For the queries about "forbidden", 99.7% can be correctly recognized, because the model only needs to complete textual reasoning. However, our task requires

Table 7: Qualitative results of the Fruit-ATVC dataset.

| Visual (V) | Text (T) | Re-created (M) | Answer (A) | **FM-Score** |
|---|---|---|---|---|
|  | Please remove the plate. |  | No problem. | **1.0** |
|  | Please put the lemon in the bowl. |  | No problem. | **0.75** |
|  | Please exchange the position of the banana and the coconut. | / | This action cannot be done. Because there is no coconut. | **1.0** |
|  | Please remove the bottle. | / | This action cannot be done. Because there is no kiwi and no bottle. | **0.75** |

Table 8: Image re-creation evaluation on the Fruit-ATVC dataset.

| PSNR | SSIM | FSIM | Human Rank (%) | | | | |
|---|---|---|---|---|---|---|---|
| | | | A | B | C | Score | FM-Score |
| 44.1 | 0.9272 | 0.420 | 12.8 | 29.4 | 57.9 | 27.5 | 46.1 |

the model to further explain why the instruction cannot be executed, which requires the model to complete both visual and textual reasoning. Therefore, only 58.2% can correctly explain the reasons. The score is the weighted average of the above results. As there are numerous possible answers, a random guessing result is less than 1%, and thus it should be safe to conclude that the proposed method has shown impressive results.

Example qualitative results are shown in Table 4. We can conclude that the model can cay "No" to instructions and give accurate language-based explanations. In addition, it is worth mentioning that the model also implicitly learns the depth of the image and size of the objects, and thus it knows whether the object is occluded and how much it is occluded when you manipulate some objects.

### 4.3.2 Results on Fruit-ATVC

For the Fruit-ATVC dataset, the re-created results are shown in Table 8. The final score for re-creation is 46.1%, which results can produce both correct re-creation as well as the textual feedback. These results also demonstrate the the difficulty of the task, especially in real scenarios.

The quantitative results of answers are shown in Table 9. We can see that 78.3% of "can" queries can be answered correctly, and 25.0% queries about "cannot" can be correctly explained. In the "exchange position" queries, we find that although it can perform correct re-creation, the reconstruction for other objects is challenging. For the "put in" queries, the errors usually occur by creating a new target fruit without erasing it in the image. The re-creation results of the "remove" are the best among three actions. All the results suggest the challenges of this dataset of real scenes. Example qualitative results are shown in Table 7.

Table 9: Textual feedback evaluation on the Fruit-ATVC dataset.

| Can | | Cannot | | | Score |
|---|---|---|---|---|---|
| Num | Acc. | Num | Type Acc. | Exp Acc. | |
| 950 | 78.3% | 922 | 77.2% | 25.0% | 64.7% |

Table 10: Uncertainty evaluation on CLEVR-ATVC dataset.

| Visual (V) | Text (T) | Re-created (M) | Answer (A) | **Ranking** |
|---|---|---|---|---|
|  | Please exchange the color of the large blue rubber sphere and the small purple rubber cylinder. |  | No problem. | **A** |
|  | Please exchange the color of the large purple metal cube and the large red metal cylinder. |  | No problem. | **A** |
|  | Please exchange the color of the large cyan metal sphere and the small red rubber cube. |  | No problem. | **A** |
|  | Please exchange the position of the small yellow rubber cube and the small purple rubber cube. |  | No problem. | **A** |
|  | Please exchange the color of the large purple metal cube and the small blue metal cylinder. |  | No problem. | **A** |
|  | Please put the large purple metal cube under the large gray metal sphere. |  | No problem. | **B** |

### 4.3.3 Uncertainty of Image Re-creation

In this section, we would like to explore whether the model can handle the uncertainty appeared in the query. The results show that our method deals with emerging uncertainties well. As shown in Table 10, all the image re-creation results get ranking A, except for the result of the last row. The score B is due to an object being erased during image reconstruction, not due to uncertainty. For example, as shown in the fourth row of Table 10, our model only exchanges the position of one "small yellow rubber cube", which is consistent with the settings in our constructed dataset. The "exchange position" only needs to process one

Table 11: Evaluation results on the CLEVR-ATVC sub-dataset.

| Methods | Text | Image | PSNR | SSIM | FSIM | Human Rank (%) | | | |
|---|---|---|---|---|---|---|---|---|---|
| | | | | | | A | B | C | Score |
| Ours w/ VQVAE | Short | $128 \times 128$ | 56.3 | 0.9957 | 0.689 | 65.9 | 14.4 | **19.7** | 71.9 |
| Ours w/ VQVAE | Long | $128 \times 128$ | **56.6** | **0.9963** | **0.693** | **66.4** | **14.8** | 18.8 | **73.8** |

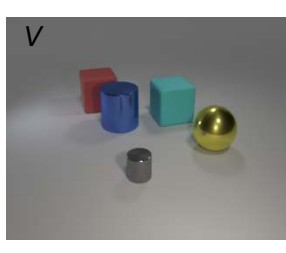

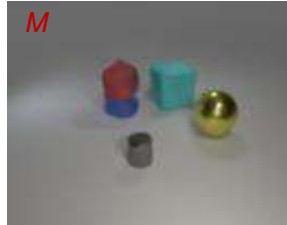

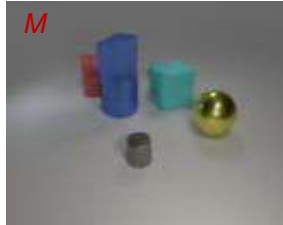

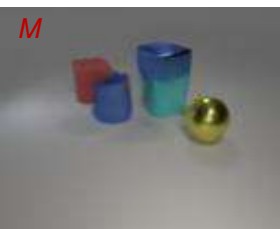

(a) Please put the big red cube on top of the big ~~blue~~ cylinder.

(b) Please put the big blue cube on top of the big blue cylinder.

(c) Please put the big blue cube on top of the big cyan cylinder.

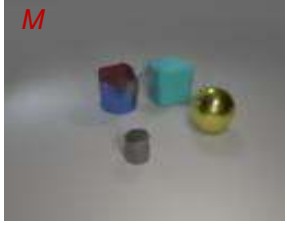

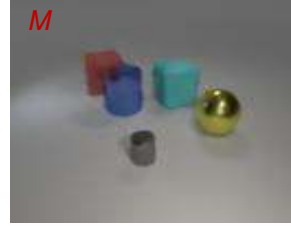

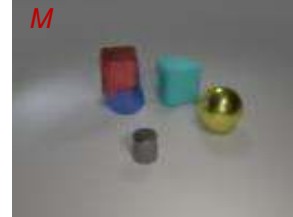

(d) Please put the big red ~~cube~~ on top of the big blue cylinder.

(e) ~~Please put~~ the big red cube on top of the big blue cylinder.

(f) Please put the amazing big red cube on top of the amazing big blue cylinder.

Figure 4: Image re-creation results on the CLEVR-ATVC sub-dataset using imperfect query. The red text represents the short or imperfect query.

of the objects, but "exchange color" needs to exchange the colors of all satisfied objects. Existing models are able to make different decisions for the above different situations, so our method shows a good behavior.

### 4.4 Single Action without Answer

In this section, we firstly evaluate the image auto-encoder method VQVAE (Oord et al., 2017) with short query and the imperfect query on the CLEVR-ATVC sub-dataset. We only select one "put on top" action without textual feedback for verification and evaluation. Subsequently, we evaluate the image auto-encoder methods, VQVAE (Oord et al., 2017) and VQGAN (Esser et al., 2021), on the CLEVR-ATVC sub-dataset.

### 4.4.1 Short or Imperfect Query for Re-creation

Qian et al. (2021) concluded that the model can produce entity memorization on the form of the input, so we test the model's ability to cope with changes in the input form. The short query means that we remove redundant text in the query, such as "please", "put", and "the" are removed, while the latter randomly removes keywords (such as "color", "size" or "shape"). The quantitative results shown in Table 11 show that the intact queries (long text) perform slightly better than the changed version (short text). This demonstrates that the model can cope with small changes in the form of the input queries.

The visualization results are shown in Figure 4, with the observations on the imperfect queries: a) If the "blue" is removed from query, the generated cylinder seems to be shorter; b) If the query contains a nonexistent "blue" adjective for the cube, the model creates one; c) If both the nonexistent "blue" cube

Table 12: Evaluation results on the CLEVR-ATVC sub-dataset.

| Methods | Image | PSNR | SSIM | FSIM | Human Rank (%) | | | |
|---|---|---|---|---|---|---|---|---|
| | | | | | A | B | C | Score |
| Ours w/ VQVAE | $128 \times 128$ | **56.6** | **0.9963** | **0.693** | **66.4** | 14.8 | 18.8 | **73.8** |
| Ours w/ VQGAN | $256 \times 256$ | 53.9 | 0.9947 | 0.604 | 52.7 | **27.4** | **19.9** | 66.4 |

| Input | Re-creation | Ground truth |
|---|---|---|

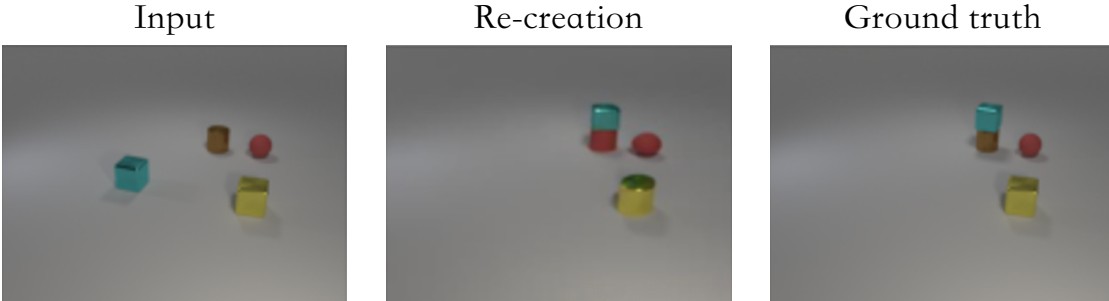

Query: Please put the small cyan cube on top of the small brown cylinder.

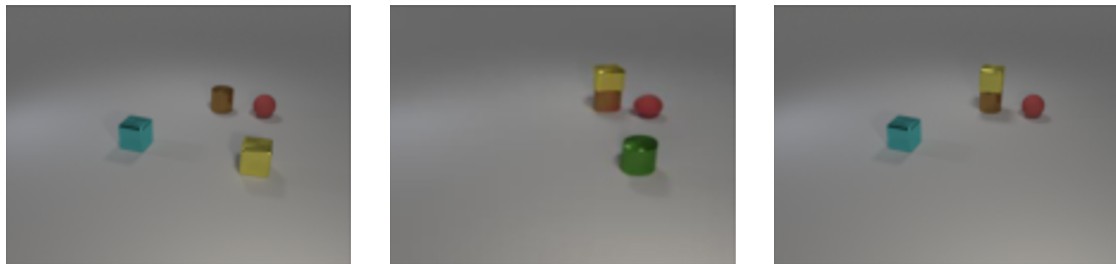

Query: Please put the small yellow cube on top of the small brown cylinder.

Figure 5: Analyse for the results of VQGAN-based image encoder. VQGAN frequently changes the colors of the objects, which downgrades the performance.

and "cyan" cylinder coexist in the same query, the model seems to mistakenly recognize the cyan cube, and we can find the small gray cylinder is also erased, which is incorrect; d) If we simply remove the "cube", the model just dyes a litter red on top of the cylinder; e) If we remove the "please put", the result just remains the original figure; f) If we add the unseen word, *e.g.*, "amazing", in the query, the manipulation result can still be achieved. We further test a lot of images, which show that the unlearned "create" and "dye" abilities commonly appear but not always.

The aforementioned findings also indicate that without incorporating restriction rules into the dataset, the model may fulfill human instructions by generating new objects, which is not the intended outcome and lacks control. Hence, it becomes imperative to govern the model's behavior by incorporating pre-defined rules as supervision signals.

### 4.4.2 VQVAE vs. VQGAN

As shown in Table 12, although VQGAN-based image auto-encoder can output a large resolution and at 4 times faster of training than the VQVAE counterpart, the latter is 7.4% better. The results shown in Figure 5 and B4 show that VQGAN-based image encoder always changes the color of objects, erases objects and add irrelevant objects on the re-created images. These results also demonstrate that VQGAN can produce high-quality images using fewer computational resources, but it appears to have lost its ability to discriminate on object properties in latent space.

# 5 Related works

In this section, we first review some representative datasets and methods on vision-and-language tasks whose inputs and outputs are always unimodal. Next, we introduce the prior multimodal dialogue models. Finally, we briefly summarize the recent controllable text-to-image generation methods and the main differences between them and our task.

## 5.1 Vision-and-Language Tasks

Recent years have witnessed the rapid development of vision-and-language tasks. Their development trend can be observed through the construction of the datasets, which can be roughly categorized into two groups: *Vision to Language*, *Language to Vision*. We briefly summarize these relevant datasets and methods.

**Vision to Language.** There are several sub-tasks for this category such as *Visual Description Generation* (Ordonez et al., 2011), *Visual Storytelling* (Lin et al., 2014), *Visual Dialog* (Das et al., 2017; Kottur et al., 2019), *Visual Entailment* (Vu et al., 2018; Liu et al., 2020), *Visual Reasoning* (Johnson et al., 2017), *Visual Question Answering* (Antol et al., 2015; Marino et al., 2019) and *Visual Referring Expression* (Liu et al., 2019).

- Visual Description Generation, a.k.a., captioning, is the most well-known task, where many datasets in the past decade are constructed to address different scales, scenes, etc. For examples, image captioning datasets, SBU1M (Ordonez et al., 2011), Flickr8k (Hodosh et al., 2013), Flickr30k (Young et al., 2014), MSCOCO (Lin et al., 2014), Multi30k-CLID (Elliott et al., 2016), Flickr30k-Entities (Plummer et al., 2015), STAIR Captions (Yoshikawa et al., 2017), Conceptual Captions (Sharma et al., 2018), MSCOCO-Entities (Cornia et al., 2019), Personality Captions (Shuster et al., 2019) and video captioning datasets, MSVD (Chen & Dolan, 2011), MPII Cooking (Rohrbach et al., 2012), YouCook (Das et al., 2013), TACoS (Regneri et al., 2013), TACoS-MultiLevel (Rohrbach et al., 2014), M-VAD (Torabi et al., 2015), MSR-VTT (Xu et al., 2016), VTW (Zeng et al., 2016), ANetCap (Krishna et al., 2017), YouCook II (Zhou et al., 2018), ANetEntities (Zhou et al., 2019), COIN (Tang et al., 2019), HowTo100M (Miech et al., 2019), have played an important role in advancing visual captioning.

- Visual Storytelling usually requires the methods to generate a series of text to describe a set or a sequence of the visual inputs, as shown in several datasets: NYC-Storytelling (Park & Kim, 2015), Disneyland Storytelling (Kim et al., 2015), SIND, and VIST (Huang et al., 2016).

- Visual Dialog can be regarded as a natural conversation system based on the image or video. Examples can be found in the existing benchmarks such as VisDial (Das et al., 2017), CLEVR-Dialog (Kottur et al., 2019) and AVSD (Alamri et al., 2018).

- Visual Entailment is introduced in V-SNLI (Vu et al., 2018), SNLI-VE (Xie et al., 2019), and VIOLIN (Liu et al., 2020), which requires the method to learn to select the correct premise and hidden semantic information that can be inferred from the visual contents.

- Vision Reasoning is expected to provide the scene graphs between objects and then to answer questions based on the visual input and relationships. Many datasets such as CLEVR (Johnson et al., 2017), NLVR (Suhr et al., 2017), CLEVR-CoGenT (Yang et al., 2018), NLVR2 (Suhr et al., 2018), GQA (Hudson & Manning, 2019), VCR (Zellers et al., 2019), and Visual COMET (Park et al., 2020) have been constructed between 2017 and 2020, which demonstrate visual reasoning abilities can be effectively learned by the deep neural networks (LeCun et al., 2015).

- Visual Question Answering is another representative vision-to-language task, in which the model is expected to generate text-based answer according to the question and the visual cues. The most influential dataset VQA v1.0 (Antol et al., 2015) as well as MovieQA (Tapaswi et al., 2016), TVQA (Lei et al., 2018), OK-VQA (Marino et al., 2019), KVQA (Shah et al., 2019), and VQA v2.0. (Antol et al., 2015) have greatly promoted the development.

- Visual Referring Expression is another task which requires the method to comprehend the referring expressions by showing evidence in the visual images, *i.e.*, using bounding boxes to locate the corresponding objects. There are several datasets targeting this task, such as RefCLEF (Kazemzadeh et al., 2014), and CLEVR-Ref+4 (Liu et al., 2019).

**Language to Vision.** This task aims at image generation based on the pure natural language. Currently, there are only a few datasets for this task such as Oxford-102 (Reed et al., 2016), Caltech-UCSD Birds (CUB) (Reed et al., 2016), Multi-Modal-CelebA-HQ (Xia et al., 2021), Text2Human (Jiang et al., 2022), Laion-400M (Schuhmann et al., 2021), Laion-5B (Schuhmann et al., 2022) and Text2Video (Xu et al., 2016; Li et al., 2018; Hong et al., 2022; Ho et al., 2022; Khachatryan et al., 2023; **?**). These datasets are similar to the vision-to-language task above, and the text-image pairs contain text descriptions of the image content. The most representative task is text-based image generation, which has recently seen large progress in multimodal learning (Brooks et al., 2023; Li et al., 2023b). Many previous works train GANs with text-conditioning on publicly available image captioning datasets (Tao et al., 2020; Zhang et al., 2021). DALL·E (Ramesh et al., 2021b) uses 250 million text-image pairs to successfully achieve promising results, which can even perform zero-shot visual reasoning by using some prompts. Another zero-shot model termed CLIP (Radford et al., 2021) is used to rank and measure the similarity between image and text, as usually one text prompt can produce numerous plausible results. The recent DALL·E 2 (Ramesh et al., 2022) uses a diffusion prior on CLIP text latents and cascaded diffusion models to generate high resolution $1024 \times 1024$ images. GLIDE (Nichol et al., 2021) also applies guided diffusion to the problem of text-conditional image synthesis. Imagen (Saharia et al., 2022) combines the power of transformer language models with high-fidelity diffusion models to deliver an unprecedented degree of photorealism in text-to-image synthesis.

In contrast to the aforementioned datasets that typically concentrate on one aspect, our proposed datasets introduce a novel requirement for models to generate both visual re-creations and textual feedback simultaneously. The visual manipulation aspect necessitates not only performing the required actions accurately but also preserving a visually plausible background. Furthermore, the feedback generated by the model needs to be aware of the possibilities of feasible actions, actions that cannot be performed, and actions that are explicitly prohibited. This requirement ensures that our constructed datasets possess both intrinsic value and present a significant challenge.

## 5.2  Multimodal Dialogue Models

The task of vision-to-language involves using visual samples as input and generating text as output, as seen in tasks like Visual Dialog. Conversely, language-to-vision tasks operate in the opposite direction. In multimodal dialogue methods, the model must reason about multimodal or unimodal inputs and produce multimodal or unimodal outputs. These methods can be categorized into the following two types[§]:

- *Multimodal Input and Unimodal Output (MIUO)*: This conversational task is similar to visual dialog, where the input typically consists of visual data and text-based prompts (i.e., multimodal input). The visual language model performs visual reasoning on the image and provides an answer (i.e., unimodal output) (Alayrac et al., 2022; Brooks et al., 2023; Gong et al., 2023; Bo Li, 2023; Li et al., 2023a; OpenAI, 2023b; Su et al., 2023; Yang et al., 2023b; Zhao et al., 2023; Zhu et al., 2023; Liu et al., 2023; Mu et al., 2023; Wang et al., 2023; Zhang et al., 2023). GPT-4 (OpenAI, 2023b) and MultiModal-GPT (Gong et al., 2023) can handle prompts consisting of interleaved visual inputs and text-based queries, generating text outputs. VideoChat (Li et al., 2023a) introduces a video-centric multimodal dialogue dataset, enabling trained models to understand and generate detailed conversations about videos.

- *Multimodal Input and Multimodal Output (MIMO)*: This task requires the model to perform multi-modal reasoning and generation simultaneously (Koh et al., 2023; Jing Yu Koh, 2023; Chenfei Wu & Duan, 2023; Yang et al., 2023a). Visual ChatGPT (Chenfei Wu & Duan, 2023) is a pioneering work that combines ChatGPT and a series of pre-trained visual foundation models, allowing them

---

[§]https://github.com/zzw-zwzhang/Awesome-of-Multimodal-Dialogue-Models

to accept and produce text and images during textual-visual conversations. GILL (Koh et al., 2023) proposes a mapping network that efficiently maps the output embedding space of a frozen text-only language model to that of a frozen generation model (e.g., Stable Diffusion (Rombach et al., 2022)). This mapping only requires fine-tuning a small number of parameters on image-caption pairs for tasks such as image retrieval, novel image generation, and multimodal dialogue. FROMAGe (Jing Yu Koh, 2023) also involves image-text inputs and outputs for multimodal dialogue, with a few linear layers fine-tuned while keeping the pre-trained language model frozen. GPT4Tools (Yang et al., 2023a) introduces an instruction dataset and extends Visual ChatGPT (Chenfei Wu & Duan, 2023) to the image understanding task.

### 5.3 Controllable Text-to-image Generation

The above text-based image generation methods primarily focus on generating high-quality images based on given text descriptions, without providing the user with the ability to manipulate specific visual attributes using natural language instructions (Anderson et al., 2018; Nguyen et al., 2019; Thomason et al., 2020; Cheng et al., 2014; Scalise et al., 2018; Stepputtis et al., 2020; Nam et al., 2018; Yüksel et al., 2021; Shen et al., 2020; Li et al., 2020; Richardson et al., 2020; Esser et al., 2021). However, some recent works have explored techniques to control the synthesis process by representing scenes as compositional generative neural feature fields, enabling the disentanglement of objects from the background, as well as individual objects' shapes and appearances (Li et al., 2019; Bodla et al., 2018; Nam et al., 2018; Chen et al., 2021; Shuster et al., 2019). For instance, ControlGAN (Li et al., 2019) can synthesize high-quality images while allowing control over specific parts of the image generation based on natural language descriptions. FusedGAN (Bodla et al., 2018) achieves enhanced controllability in sampling by disentangling different factors in the latent space. Text2Human (Jiang et al., 2022) can synthesize human images by specifying clothing shapes and textures solely through natural language descriptions.

However, the aforementioned approaches focus on controlling the image generation process through composition or disentanglement techniques. In this paper, our objective is to control the outcomes of re-created images, where the system learns to respond with a "no" to commands that are either prohibited or cannot be executed.

## 6 Conclusion

In this paper, we raise an important yet underexplored concern regarding the accountability of multimodal generative models. To tackle this issue, we introduce two novel datasets, namely CLEVR-ATVC and Fruit-ATVC, designed for a unique task called Accountable Text-based Visual Re-creation. The objective of this task is to train Visual Language Models (VLMs) to reject human instructions. Our datasets consist of both visual and textual inputs and outputs, requiring the model to perform visual re-creation while ensuring image quality when the answer is "yes". In cases where the model cannot complete the action or the action is prohibited, it is required to provide an explanation. We provide baseline models, experimental settings, evaluation metrics, and a comprehensive analysis, presenting some promising results. These high-quality datasets can also be utilized in similar tasks, and we hope that our work inspires further research on the accountability problem, encompassing model design, label annotation, and the creation of larger datasets. We firmly believe that addressing the issue of accountability is crucial for advancing the development and deployment of multimodal generative models.

## 7 Acknowledgments

The authors would like to thank the anonymous reviewers for their helpful feedback and the funding agencies listed below for supporting this work. This work is supported in part by the National Key Research and Development Program of China (Project Number 2022YFC2305102). We would like to thank Ying Cai, Xiaoai Sha and other members for helping label text-image pairs and participate in human evaluation experiments. Finally, we appreciate all the people who are working on making code, models and data publicly available.

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

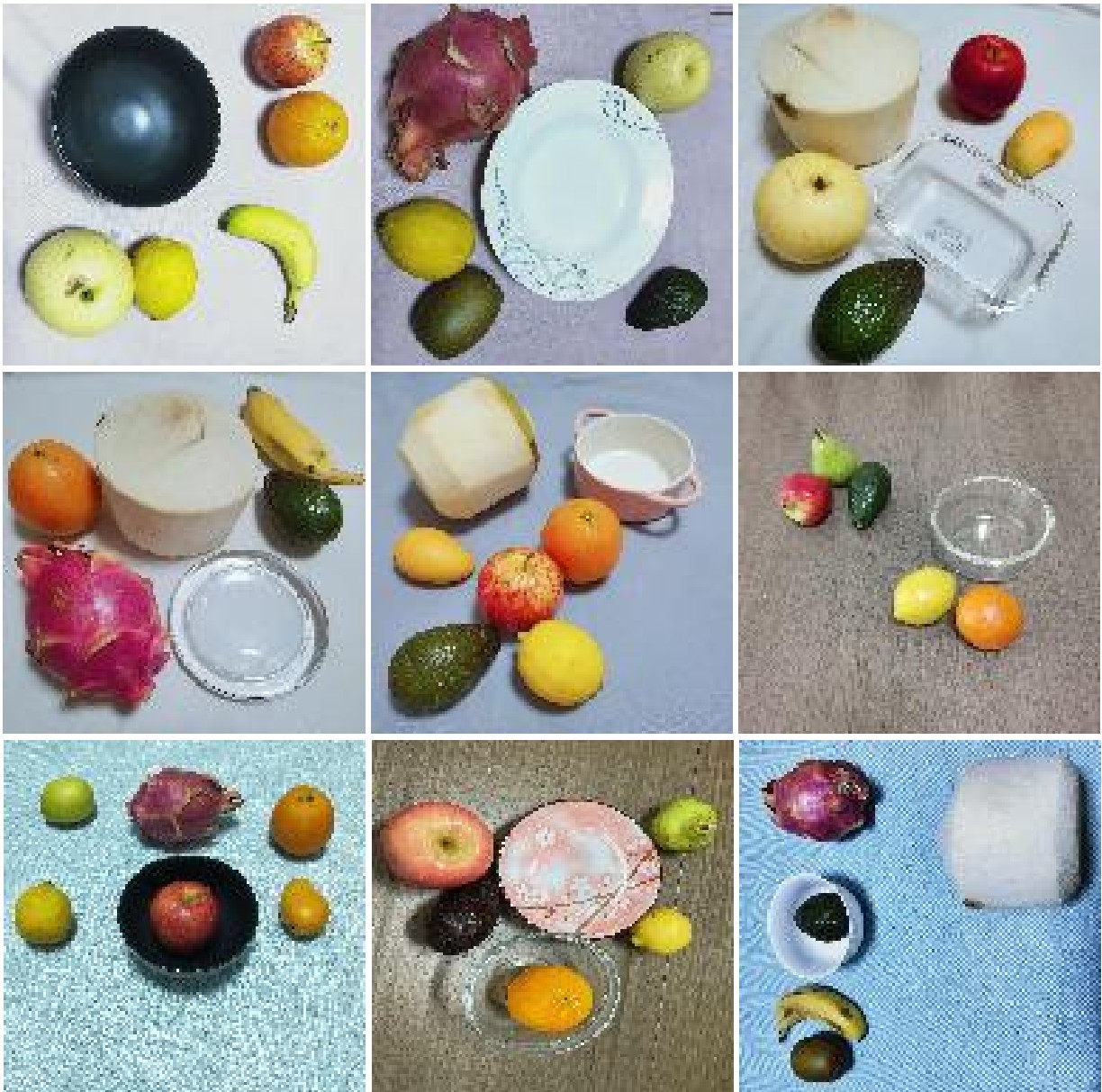

Figure A1: The example images of Fruit-ATVC dataset show the diversity of collection scenes.

# A    Additional Data Details

This section provides further information about the data used in our research. We begin by presenting a wider range of example images from the Fruit-ATVC dataset to highlight the diversity of collection scenes. Next, we provide a summary of statistics regarding the various actions found in the datasets. Subsequently, we elaborate on the process of human evaluation experiments. Lastly, we introduce the format of the annotation files for our datasets.

## A.1    Diversity of Fruit-ATVC

The following Figure A1 shows the diversity of collection scenes on Fruit-ATVC dataset. The image resolution is average 2700×2700, and the images were taken with different categories of mobile phones, including iphone,

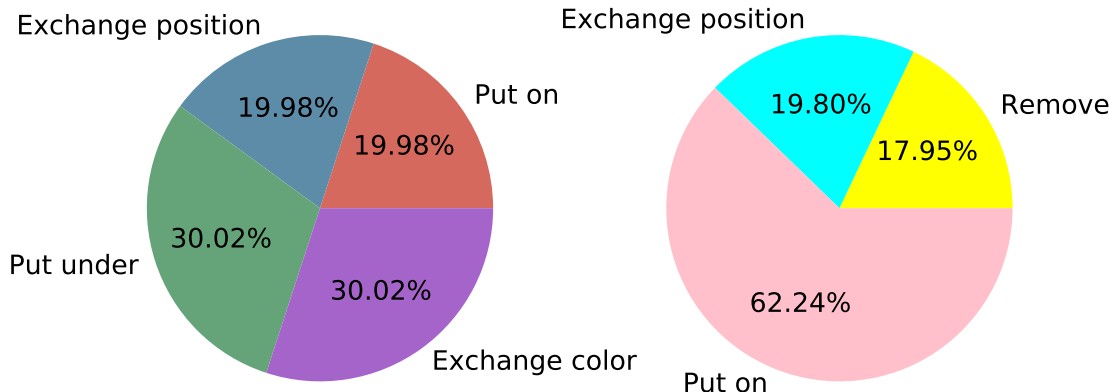

Figure A2: The distribution of different actions on CLEVR-ATVC and Fruit-ATVC datasets.

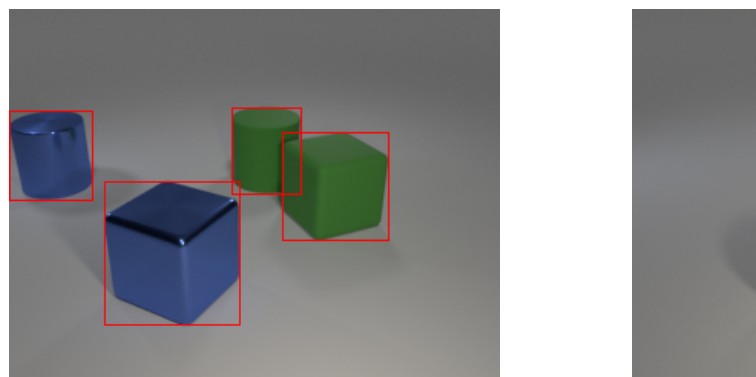
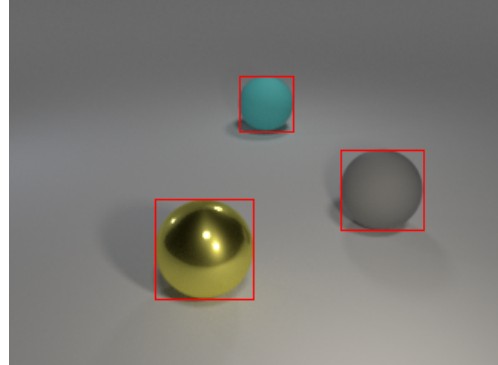

Figure A3: The visualization of bounding boxes in the CLEVR-ATVC dataset.

huawei, xiaomi, etc. We would like to emphasize that the image quality of this dataset is very high, which results in the size of the original Fruit-ATVC is 168 GB. In addition, the background of data collection, fruit categories and container types are also very diverse. We believe this dataset can also be used for other image generation tasks. The Figure A2 illustrates the distribution of different actions on our constructed datasets.

## A.2 Annotation File Format

We describe the format of the annotation files for both the CLEVR-ATVC and Fruit-ATVC datasets. Listings 1 and 2 demonstrate the consistent structure of the annotation files, with the primary distinction being the inclusion of detailed 2D and 3D coordinate locations of objects in the CLEVR-ATVC dataset. Figure A3 provides a visual representation of the bounding boxes in the CLEVR-ATVC dataset. It is worth noting that we anticipate the potential use of this dataset for other related tasks, such as compositional visual reasoning. While we have previously introduced the main structure of the annotation files in the main paper, we now aim to provide additional details.

The annotation files begin by providing a summary of the dataset contributors, timestamp, project link, version, and license information. This introductory section ensures comprehensive documentation and proper attribution. Following this, the annotation files include the categories of objects present in the dataset, along with associated object adjectives, actions, and other relevant information.

The "images" section contains all the necessary information about the visual inputs. This includes the image index, filename, link, the number of objects in the image, and detailed descriptions of each object present. Each object is listed under the "objects" child list, accompanied by specific properties such as index, category index, and bounding box coordinates.

The text-based queries (Q) and textual feedback (A) are included in the "questions" section of the annotation files. The answer types are indicated by numbers ranging from 0 to 2. Specifically, a value of 0, 1, or 2 represents an instruction that cannot be executed, an instruction that can be executed, or a prohibited instruction, respectively. If the action type is 0 or 2, the subsequent "actions" list in the "recreations" section will be empty. Notably, the CLEVR-ATVC dataset provides additional information regarding any changes in the color or position of an object after a displacement or property modification.

## B  Human Evaluation Experiments

This section presents examples of the interfaces utilized for facilitating human evaluation. During the evaluation phase, the visual input, text-based query, re-created image, answer, and ground truths for re-creation and textual feedback are automatically imported into the corresponding sections of the auxiliary tool. Evaluators are then only required to assign scores to the respective results and save them. Subsequently, we calculate the final score of the model based on the saved results.

As detailed in the main paper, we assigned the same results to different evaluators, with each evaluator providing identical scores for the experimental outcomes. The aforementioned experiments demonstrate that the auxiliary tools we have designed not only significantly reduce the evaluation time required but also ensure the accuracy and reliability of the widely used human evaluation process.

Listing 1: The format of annotation file on CLEVR-ATVC dataset.

```
{
    "contributor": "Zhiwei Zhang, Yuliang Liu",
    "data_created": "2023",
    "url": "https://matrix-alpha.github.io/",
    "description": "CLEVR-ATVC Dataset",
    "vertion": "1.0",
    "licenses": {
        "id": "1",
        "url": "Nnone",
        "name": "Creative Commons Attribution (CC-BY 4.0)",
    },
    "categories": ["cube, cylinder, sphere"],
    "sizes": ["small, large"],
    "colors": ["gray, red, blue, green, brown, purple, cyan, yellow"],
    "material": ["rubber, metal"],
    "actions": ["put on top, put under, exchange position, exchange color"],
    "images": [
            {
            "image_idx": "0000001",
            "image_filename": "521100_03_000011.png",
            "id": "521100_03_000011",
            "object_number": 3,
            "data_created": "2023-04-27 00:17:14",
            "objects": [
                {
                "index": 0,
                "category_id": 0,
                "size_id": 1,
                "color_id": 3,
                "bbox": [29, 62, 81, 116],
                "3d_coords": [-2.96856045, -1.99158716, 0.6999999],
                "material_id": 1},
                {...},
                {...},
                        ]
            },
        ]
    "questions": [
            {
            "question_idx": "0000001",
            "question_id": "521100_03_000011",
            "question_number": 10,
            "ques": [
                {
                "ques_id": 0,
                "ques_idx": 1,
                "id": "521100_03_000011_0_01",
                "type": 1,
                "Q": ["Please put the cylinder on top of the cube."],
                "A": ["No problem."]
                },
            ]
        "recreations": [
```

```
{
"rec_idx": "0000001",
"rec_id": "521100_03_000011",
"rec_num": 10,
"actions": [
    {
    "actions_id": 0,
    "actions_idx": 1,
    "rec_filename": "521100_03_000011_0_01.png",
    "object_number": 3,
    "date_created": "2023-04-27 00:17:59",
    "objects": [
    {},
    {},
    {
    "index": 2,
    "category_id": 1,
    "size_id": 0,
    "color_id": 1,
    "bbox": [39, 45, 65, 72],
    "3d_coords": [-2.96856045, -1.99158716, 1.75],
    "material_id": 0}
                ]
    },
]
}
```

Listing 2: The format of annotation file on Fruit-ATVC dataset.

```
{
  "contributor": "Zhiwei Zhang, Yuliang Liu",
  "data_created": "2023",
  "url": "https://matrix-alpha.github.io/",
  "description": "Fruit-ATVC Dataset",
  "vertion": "1.0",
  "licenses": {
    "id": "1",
    "url": "Nnone",
    "name": "Creative Commons Attribution (CC-BY 4.0)",
  },
  "fruits": ["apple, cocount, orange, banana, kiwi, mango, avocado, etc."],
  "containers": ["plate, bowl, bottle"],
  "actions": ["put in, exchange position, remove"],
  "images": [
        {
        "image_idx": "0000001",
        "image_filename": "0000001.png",
        "id": "0000001",
        "fruit_number": 3,
        "container_number": 1,
        "data_created": "2023-04-27 00:10:11",
        "fruits": [
                {
                    "index": 0,
                    "category_id": 0
                }
                {
                    "index": 1,
                    "category_id": 6
                }
                {
                    "index": 2,
                    "category_id": 8
                }
            ],
        "containers": [
                {
                    "index": 0,
                    "category_id": 2
                }
            ],
        },
    ]
  "questions": [
        {
        "question_idx": "0000001",
        "question_id": "0000001",
        "question_number": 3,
        "ques": [
            {
            "ques_id": 0,
            "ques_idx": 1,
```

```
                "id": "0000001_0_01",
                "type": 1,
                "Q": ["Please put the kiwi in the plate."],
                "A": ["No problem."]
                },
        ]
    "recreations": [
        {
        "rec_idx": "0000001",
        "rec_id": "0000001",
        "rec_num": 3,
        "actions": [
                {
                "actions_id": 0,
                "actions_idx": 1,
                "rec_filename": "0000001_0_01.png",
                "fruit_number": 3,
                "container_number": 1,
                "date_created": "2023−04−27 00:10:18",
                },
        ]
}
```

Input        Re-creation        Ground truth

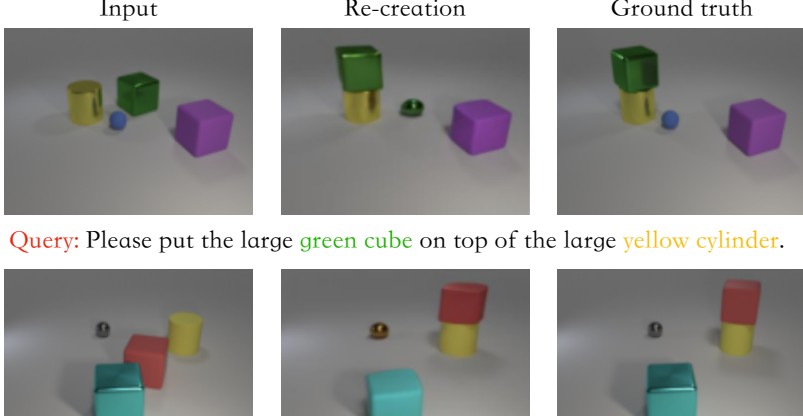

Query: Please put the large green cube on top of the large yellow cylinder.

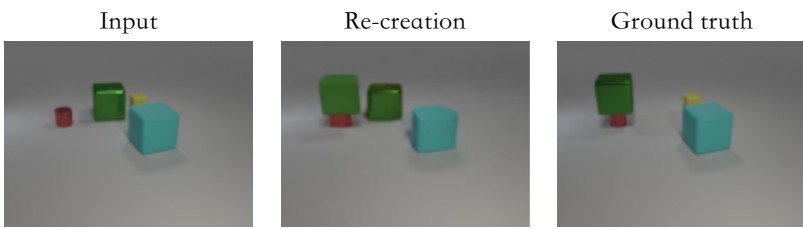

Query: Please put the small red cube on top of the large yellow cylinder.

Input        Re-creation        Ground truth

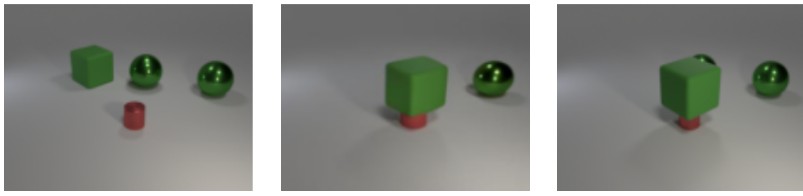

Query: Please put the large green cube on top of the small red cylinder.

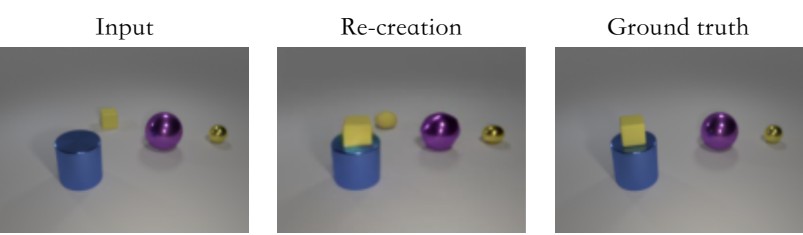

Query: Please put the large green cube on top of the small red cylinder.

Input        Re-creation        Ground truth

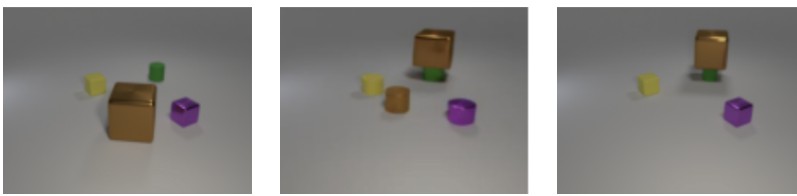

Query: Please put the small yellow cube on top of the large blue cylinder.

Query: Please put the large brown cube on top of the small green cylinder.

Figure B4: More results are to show why VQGAN-based image encoder performs worse on image re-creation.

|  Input | Re-creation | Ground truth |

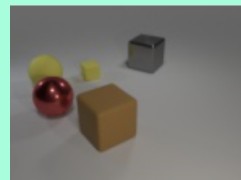 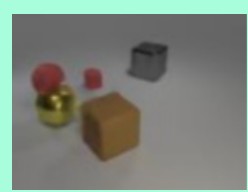 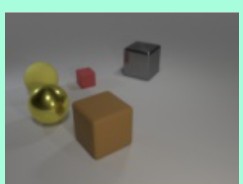

*Query:* Please exchange the color of the small yellow rubber cube and the large red metal sphere.

*Answer:* No problem.

*Ground truth:* No problem.

### Human Evaluation

1. Please rate the visual re-creation?

    ● A: 1      ○ B: 0.5      ○ C: 0

2. Please rate the type of answer?

    ● 1      ○ 0

3. Please rate the explanation of answer?

    ● 1      ○ 0

4. What is the score for textual feedback? [1.0]

5. What is the full-match score? [1.0]

|  Input | Re-creation | Ground truth |

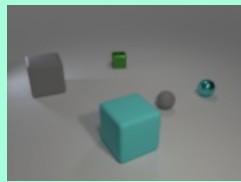

*Query:* Please put the large gray metal cylinder under the small yellow rubber cylinder.

*Answer:* This action cannot be done. Because there is no large gray metal cylinder and no small yellow rubber cylinder.

*Ground truth:* This action cannot be done. Because there is no large gray metal cylinder and no small yellow rubber cylinder.

### Human Evaluation

1. Please rate the visual re-creation?

    ● A: 1      ○ B: 0.5      ○ C: 0

2. Please rate the type of answer?

    ● 1      ○ 0

3. Please rate the explanation of answer?

    ● 1      ○ 0

4. What is the score for textual feedback? [1.0]

5. What is the full-match score? [1.0]

|  Input | Re-creation | Ground truth |
| --- | --- | --- |

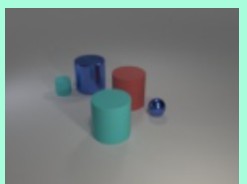

*Query:* Please put the large tray metal cube on top of the large blue metal sphere.

*Answer:* This action is forbidden. Because you cannot put an object on top of the sphere, and there is no large gray metal cube and no large blue metal sphere.

*Ground truth:* This action is forbidden. Because you cannot put an object on top of the sphere, and there is no large gray metal cube and no large blue metal sphere.

### Human Evaluation

1. Please rate the visual re-creation?

    ● A: 1        ○ B: 0.5        ○ C: 0

2. Please rate the type of answer?

    ● 1        ○ 0

3. Please rate the explanation of answer?

    ● 1        ○ 0

4. What is the score for for textual feedback? [1.0]

5. What is the full-match score? [1.0]

|  Input | Re-creation | Ground truth |
| --- | --- | --- |

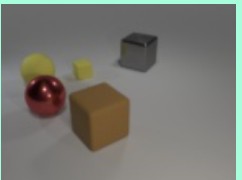

*Query:* Please put the large red metal sphere under the small yellow rubber cube.

*Answer:* This action is forbidden. Because you cannot put the sphere under an object, and there is no large red metal sphere and no small yellow rubber cube.

*Ground truth:* This action is forbidden. Because you cannot put the sphere under an object.

### Human Evaluation

1. Please rate the visual re-creation?

    ● A: 1        ○ B: 0.5        ○ C: 0

2. Please rate the type of answer?

    ● 1        ○ 0

3. Please rate the explanation of answer?

    ○ 1        ● 0

4. What is the score for for textual feedback? [0.5]

5. What is the full-match score? [0.75]

| Input | Re-creation | Ground truth |
|---|---|---|

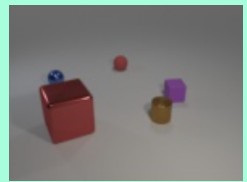

*Query:* Please put the small yellow metal sphere under the large red metal cube.

*Answer:* This action is forbidden. Because you cannot put the sphere under an object, and there is no small yellow metal sphere and no small red metal cube.

*Ground truth:* This action is forbidden. Because you cannot put the sphere under an object, and there is no small yellow metal sphere and no large red metal cube.

### Human Evaluation

1. Please rate the visual re-creation?

    ● A: 1     ○ B: 0.5     ○ C: 0

2. Please rate the type of answer?

    ● 1     ○ 0

3. Please rate the explanation of answer?

    ○ 1     ● 0

4. What is the score for for textual feedback? [0.5]

5. What is the full-match score? [0.75]

| Input | Re-creation | Ground truth |
|---|---|---|

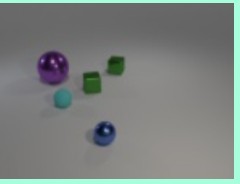 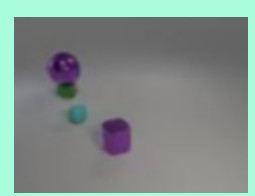 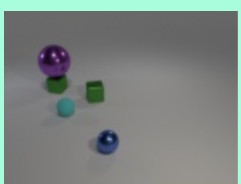

*Query:* Please put the small green metal cube under the large purple metal sphere.

*Answer:* No problem.

*Ground truth:* No problem

### Human Evaluation

1. Please rate the visual re-creation?

    ○ A: 1     ● B: 0.5     ○ C: 0

2. Please rate the type of answer?

    ● 1     ○ 0

3. Please rate the explanation of answer?

    ● 1     ○ 0

4. What is the score for for textual feedback? [1.0]

5. What is the full-match score? [0.75]

| Input | Re-creation | Ground truth |
|:---:|:---:|:---:|
| 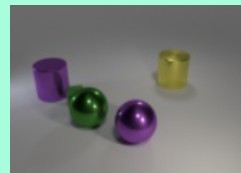 | 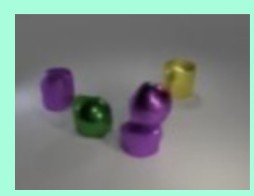 | 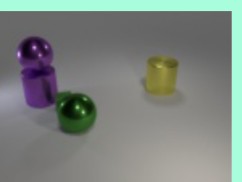 |

*Query:* Please put the large purple metal sphere on top of the large purple metal cylinder.

*Answer:* This action is forbidden. Because you cannot put the sphere under an object.

*Ground truth:* No problem.

### Human Evaluation

1. Please rate the visual re-creation?

    ○ A: 1         ○ B: 0.5         ● C: 0

2. Please rate the type of answer?

    ○ 1         ● 0

3. Please rate the explanation of answer?

    ○ 1         ● 0

4. What is the score for for textual feedback?  [0]

5. What is the full-match score?  [0]

---

| Input | Re-creation | Ground truth |
|:---:|:---:|:---:|
| 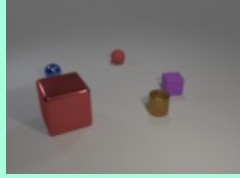 | 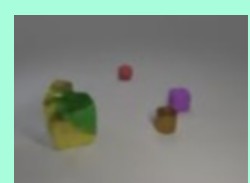 | |

*Query:* Please exchange the position of the large yellow metal cube and the large green metal cylinder.

*Answer:* No problem.

*Ground truth:* This action cannot be done. Because there is no large yellow metal cube and no large metal cylinder.

### Human Evaluation

1. Please rate the visual re-creation?

    ○ A: 1         ○ B: 0.5         ● C: 0

2. Please rate the type of answer?

    ○ 1         ● 0

3. Please rate the explanation of answer?

    ○ 1         ● 0

4. What is the score for for textual feedback?  [0]

5. What is the full-match score?  [0]

