# OpenReview forum: "Accountable Textual-Visual Chat Learns to Reject Human Instructions in Image Re-creation"
_TMLR — Accepted by TMLR_

### Review · Reviewer_KJPy · 2023-11-04

**Summary Of Contributions:**

This paper propose a new task ATVC - a dataset where models have to generate images and decline generations if it is not possible to do so. They create 2 datasets for this task. They train a model similar to DALL-E on their dataset and analyze their results.

**Audience:**

Yes

**Claims And Evidence:**

Yes

**Requested Changes:**

Major
- clarify of contribution as listed above

Minor
- Add GPT-4 results

Questions
- Are the answer types part of the ATVC framework? If so, it would make the paper easier to follow if the answer types are described in section 2.1
- “For example, six human instructions can be executed by the AI system and re-created the images accordingly, two instructions cannot be executed and two instructions are forbidden.” - Is the number of queries for each action type is randomly chosen for each image or always used the breakdown described in the example?
- What is the difference between an instruction that is forbidden and an instruction that cannot be executed? An instruction that is forbidden also cannot be executed.
- The two types of prohibition are cannot put object under/on top of other object? Is this prohibition based on physics? And why only these prohibitions of all the prohibitions? These two prohibitions seems arbitrary.
- In the uncertainty experiments, the model handles it well by choosing one object (if multiple objects are correct). Shouldn’t the correct behavior be to clarify with the user which object is correct?
- In the imperfect query for re-creation results, does this mean the model is not robust and cannot handle noisy inputs?

**Strengths And Weaknesses:**

Strength
- The task of generating images (while possibly rejecting human instructions) is a new and impactful task that should be studied.
- The dataset is constructed to capture this task well.


Weakness
- Clarity of contribution. What distinguishes and motivates this particular multimodal dataset? From my understanding, it is that the dataset measures the quality of generated text and image. The generated text is only required since the authors introduce a new task of ATVC where the model can reject the request. If my understanding is correct, I think the paper should motivate this new task first and then the accompanying dataset and make this more clear in the intro.
- Lacking GPT-4 results, though GPT-4 is the motivation. The motivation for constructing this multimodal dataset is to evaluate the performance of multimodal generative models like ChatGPT and GPT-4. However, there are no results of GPT-4 on this dataset. I understand GPT-4 requires paid access, but given that this is the motivation for the dataset in the first place, these results should be included to justify the motivation. Also, including results on GPT-4 would make this work more impactful and useful to others.

---

### Review · Reviewer_RgT9 · 2023-12-13

**Summary Of Contributions:**

The paper points out the lack of a dataset for validating model performance in the textual-visual chat tasks and proposes two datasets specifically designed to evaluate the quality of image and text outputs in multimodal dialogue systems. The proposed datasets include CLEVR-ATVC, a synthetic dataset, and Fruit-ATVC, containing real images, both including visual and text-based inputs and outputs. For the first time in the field of multimodal dialogue systems, these datasets incorporate specific rules to enable the model to reject specific human requests.
To explore the capability of rejecting human instructions in a textual-visual chat task, the paper proposes an accountable text-based visual re-creation task using the proposed datasets. This task involves receiving text-image pairs and providing re-created images with language-based feedback including explanations for the rejection of requests in specific cases. To achieve this task, the paper suggests a two-stage training procedure; training a discrete VAE (dVAE) for compressing each image into short tokens and training a decoder-based transformer that takes the previously generated image tokens concatenated with text tokens as input for generating final outputs.
The experiments analyze the quality of the re-created images, the accuracy of responses, and the model’s behavior when handling imperfect and ambiguous user queries. Additionally, it compares the performance of VQVAE and VQGAN as image auto-encoders, revealing that VQGAN shows lower performance.

**Audience:**

Yes

**Broader Impact Concerns:**

-     It appears that there are no specific ethical issues that need to be particularly considered regarding the paper.

**Claims And Evidence:**

Yes

**Requested Changes:**

-	It seems necessary to have test set requests that use different expressions but hold similar meanings to the forbidden rules present in the training set. This would enable the evaluation of whether the model robustly recognizes forbidden expressions.

**Strengths And Weaknesses:**

[Strengths]
-	The newly proposed datasets seem to have the potential to make a meaningful impact within this research field. Their intended design, which consist of image-text inputs and outputs including the prohibited requests, aligns well with the current trends on recent generative dialogue systems. Therefore, evaluations with these datasets are expected to be utilized in future studies.

[Weaknesses]
-	Evaluating the ability to reject prohibited requests using the proposed dataset seems to be inappropriate. The exceptionally high performance of ‘Type Acc.’ (99.7%) for 'forbidden' in Table 6 suggests that recognizing prohibited requests may be a straightforward task. The test samples in the dataset may be too simplistic, diminishing the effectiveness to evaluate the challenges in rejecting prohibited requests.

-	The paper does not seem to demonstrate a clear explanation for why Fruit-ATVC, one of the proposed datasets, does not include forbidden rules. Considering the ethical implications of rejecting specific requests on real images, it appears necessary for the paper to articulate the rationale behind the absence of forbidden rules within the Fruit-ATVC dataset.

---

### Review · Reviewer_EcwY · 2023-12-20

**Summary Of Contributions:**

The paper proposes new two datasets to evaluate the multimodal generation capabilities of Visual Language Models (VLMs): the synthetic CLEVR-ATVC dataset and the manually pictured Fruit-ATVC dataset. The most notable feature of these datasets is that on these datasets, the model is required to reject textual prompts if they are prohibited according to pre-defined rules or cannot be executed. The authors develops a simple baseline model consisting of VQVAE and an autoregressive Transformer, and show quantitative and qualitative experiment results for the generated image quality and answer using the baseline. They also evaluate the model's behavior when facing uncertain or imperfect queries.

**Audience:**

Yes

**Broader Impact Concerns:**

Please add the broader impact of the work.

**Claims And Evidence:**

No

**Requested Changes:**

Please modify the draft to reflect what I mentioned above. Also, provide additional results using at lest two or three other VLMs.

**Strengths And Weaknesses:**

+ As authors mentioned in the conclusion section, the collected datasets will foster future research on the accountability of VLMs.
+ The proposed baseline seems to manipulate image input well based on the textual prompt, or reject the prompt if it cannot be executed or is prohibited.

However, the writing should be improved.
- Generation of textual queries and feedbacks on Fruit-ATVC is unclear. Please elaborate on the interface shown in Figure 2.
- Some of the sections should be differently represented; e.g., Section 4.3.3 and Section 4.4.3 describe redundant issues (VQVAE vs VQGAN), the descriptions about imperfect queries are mentioned in Section 4.4.1, not Section 4.4.2, and many examples mentioned in Section 4.4.2 are not about imperfect queries, but the queries that cannot be executed or unseen queries.
- The attention mask used in the decode-based Transformer should be described in more detail. The following sentences in Section 3 are quite unclear: The transformer is a pure decoder, where text tokens can be attended by each image token in any one of the self-attention layers. Causal mask and full attention mask are used for the text-to-text and image to-image, respectively.
- The paper addresses the accountability of VLMs, but it is limited in terms of object attributes. The VLMs must also reject prompts requiring unethical behaviors or that cannot be executed in more diverse scenarios, such as the prompt that defies the law of physics. The authors should address the accountability of VLMs in more detail in the Related Work section.

And most importantly, conducting experiments with only the proposed baseline does not fully demonstrate the model's ability and reduces the reliability of dataset analysis.

---

### Decision · Action_Editor_xcLc · 2024-02-03

**Recommendation:** Accept with minor revision

**Comment:**

The paper introduces new multimodal datasets/tasks that require models to reject human instructions that may be physically impossible. All three reviewers and the AE appreciated the novelty and importance of the task introduced, the effort in curating the dataset and tasks, and the thorough experiments to benchmark models for these tasks. The paper is recommended for conditional acceptance. I describe some of the remaining issues that need to be addressed below. Additionally, the authors did a commendable job in surveying the related work, and as such the paper is recommended for Survey certification. Congratulations to the authors!

Major comment:
- Given the limited set of actions/names/colors/etc, powerful transformers are expected to memorize these names and patterns, and it is likely that they won't generalize to unseen ones. As an example, see (Qian et al., 2021) where transformer-based dialog models do not generalize to unseen named entities. I suggest that the authors leave out some of the colors/names/actions/etc or create unseen ones for the test set only, and devise a clear recipe for testing generalization to unseen objects/actions/etc, which is one of the hallmarks of using foundation models for solving new tasks.

Qian, Kun, et al. "Annotation Inconsistency and Entity Bias in MultiWOZ." Proceedings of the 22nd Annual Meeting of the Special Interest Group on Discourse and Dialogue. 2021.

Minor comments:
- End of page 5: Figure 2.4 -> Figure 3
- Please use \𝗰𝗶𝘁𝗲𝘁{} and \𝗰𝗶𝘁𝗲𝗽{} correctly or the paper will be hard to read. \𝗰𝗶𝘁𝗲𝘁{} should be used when the author name is intended to be part of the sentence, e.g., "X et al. (2024) introduced a new benchmark." \𝗰𝗶𝘁𝗲𝗽{} should be used when the citation appears in parentheses and is not read in the sentence, e.g., "There has been great progress in vision language models (X et al., 2024)."
- Please move the tables and images around so that they appear in the same order as they appear in the text, and preferably closer to where they appear in the text.
- Please remove the page break between Conclusion and References.
- Several lines in the annotation formats in the appendix don't fit the page. Please fix them.

**Audience:**

The curated datasets are certainly of interest to the broader community.

**Claims And Evidence:**

The dataset curation has been explained well, and supports the claims of the paper.

---

> ### Author Response · Authors · 2024-02-08
> **Thanks for your recommendation!**
>
> 1) We have modified the paper based on the minor comments.
>
> 2) Response to major comment: 1) The problem you pointed out is similar to our experiment in Section 4.4.1, where we further introduced the conclusion of mentioned paper. 2) For unseen objects, our pre-trained codebook cannot re-create them with high quality. Furthermore, just as large language models have strong zero-shot learning capabilities, our current models have difficulty generalizing to unlearned instructions. Both of the above problems require larger multimodal datasets and larger models to be solved. Therefore, we think that it is too early to consider the model generalization at this time. 3) We would like to emphasize that our datasets are valuable for exploring the architectural design of multimodal generative models because its training cost is acceptable to most researchers. 4) We will continue to explore this task and hope that future multimodal foundation models will have stronger generalization, especially on the accountability issues. We hope our clarifications can solve your concerns.
>
> We thank Action Editor for recommending our paper to Survey certification. Thanks again to all the reviewers.